# The Combination of Albumin–Bilirubin Score and Prothrombin Time Is a Useful Tool for Predicting Liver Dysfunction after Transcatheter Arterial Chemoembolization in Child–Pugh Class A Patients with Hepatocellular Carcinoma within Up-to-Seven Criteria

**DOI:** 10.3390/jcm10214838

**Published:** 2021-10-21

**Authors:** Hiroaki Takaya, Tadashi Namisaki, Soichi Takeda, Kosuke Kaji, Hiroyuki Ogawa, Koji Ishida, Yuki Tsuji, Hirotetsu Takagi, Takahiro Ozutsumi, Yukihisa Fujinaga, Masanori Furukawa, Koh Kitagawa, Norihisa Nishimura, Yasuhiko Sawada, Naotaka Shimozato, Hideto Kawaratani, Kei Moriya, Takemi Akahane, Akira Mitoro, Hitoshi Yoshiji

**Affiliations:** Department of Gastroenterology, Nara Medical University, 840 Shijo-cho, Kashihara 634-8522, Nara, Japan; tadashin@naramed-u.ac.jp (T.N.); souitit@naramed-u.ac.jp (S.T.); kajik@naramed-u.ac.jp (K.K.); ogawah@naramed-u.ac.jp (H.O.); ishidak@naramed-u.ac.jp (K.I.); tsujih@naramed-u.ac.jp (Y.T.); htakagi@naramed-u.ac.jp (H.T.); ozutaka@naramed-u.ac.jp (T.O.); fujinaga@naramed-u.ac.jp (Y.F.); furukawa@naramed-u.ac.jp (M.F.); kitagawa@naramed-u.ac.jp (K.K.); nishimuran@naramed-u.ac.jp (N.N.); yasuhiko@naramed-u.ac.jp (Y.S.); shimozato@naramed-u.ac.jp (N.S.); kawara@naramed-u.ac.jp (H.K.); moriyak@naramed-u.ac.jp (K.M.); stakemi@naramed-u.ac.jp (T.A.); mitoroak@naramed-u.ac.jp (A.M.); yoshijih@naramed-u.ac.jp (H.Y.)

**Keywords:** albumin, bilirubin, therapeutic chemoembolization, hepatocellular carcinoma, liver dysfunction

## Abstract

Mortality and recurrence rates of hepatocellular carcinoma (HCC) are high. Recent studies show that for patients with HCC beyond up-to-seven criteria, treatment with molecular-targeted agents (MTAs) is recommended because the treatment efficiency of transcatheter arterial chemoembolization (TACE) is poor; further, TACE increases decline in liver function. However, the relationship between TACE and liver function decline in patients with HCC within up-to-seven criteria has not been clarified. Hence, we aimed to investigate this relationship. This retrospective observational study included 189 HCC tumors within up-to-seven criteria in 114 Child–Pugh class A patients. Twenty-four (12.7%) tumors were changed from Child–Pugh class A to B after TACE, and 116 (61.4%) tumors exhibited recurrence within 6 months after TACE. Prothrombin time (PT) and albumin–bilirubin (ALBI) score before TACE were significantly associated with liver dysfunction from Child–Pugh class A to B. The combination of PT and ALBI score before TACE had high predictive ability for liver dysfunction from Child–Pugh class A to B after TACE (specificity = 100%, sensitivity = 91.7%). The combined use of pre-TACE PT and ALBI score has a high predictive ability for liver dysfunction after TACE for Child–Pugh class A patients with HCC within up-to-seven criteria.

## 1. Introduction

Hepatocellular carcinoma (HCC) is the fourth leading cause of cancer-related death in the world [1,2,3]. The medical treatment policy in Japan is based on the consensus-based clinical practice guidelines for HCC management of the Japan Society of Hepatology (JSH). For patients with ≥4 HCCs, treatment with transcatheter arterial chemoembolization (TACE) or molecular-targeted agents (MTAs) is recommended [4]. In Japan, treatment with MTAs is not recommended for patients with HCC and Child–Pugh class B because of liver dysfunction. In other words, patients with HCC should be treated with MTAs before they experience a decline in liver function. However, some patients with HCC show a decline in liver function after undergoing TACE [5,6], and the recurrence rate of HCC after TACE is high [7]. Therefore, some patients should not be treated with MTAs for HCC recurrence.

Recent studies have reported that for patients with HCC beyond up-to-seven criteria [6,8], treatment with MTAs is recommended as the treatment efficiency of TACE for HCC beyond up-to-seven criteria is poor [6] and TACE increases the decline in liver function [5,6]. Other studies have reported that the treatment efficiency of MTAs depends on liver function [5,9].

Patients with HCC within up-to-seven criteria as well as those beyond up-to-seven criteria occasionally experience a decline in liver function from Child–Pugh class A to class B after undergoing TACE. Recent studies have reported that the prognosis of patients with HCC is associated with liver function [5,10]. In other words, to improve prognosis, the maintenance of liver function is important in patients with HCC. Because MTA treatment does not significantly affect the decline in liver function [6], it is important to investigate the relationship between TACE and decline in liver function in patients with HCC within up-to-seven criteria as well as beyond up-to-seven criteria.

The albumin–bilirubin (ALBI) score, which uses only albumin (Alb) and total bilirubin (T-Bil), is a useful tool for evaluating liver function [11,12]. Studies have reported that the ALBI score can be used instead of the Child–Pugh score [13] to predict the prognosis of patients with HCC [14,15].

In the present study, we first investigated the relationship between TACE and decline in liver function as well as risk factors for progression from Child–Pugh class A to class B after TACE in patients with HCC within up-to-seven criteria using the ALBI score. Subsequently, we studied the risk factors for early recurrence after TACE.

## 2. Materials and Methods

### 2.1. Study Design and Patients

This retrospective observational study included 361 HCC tumors in 186 patients who underwent TACE [7] at our hospital between March 2015 and August 2020. HCC was diagnosed by dynamic computed tomography (CT) and/or dynamic magnetic resonance imaging (MRI) following the JSH consensus-based clinical practice guidelines for HCC management [4]. Patient eligibility criteria were HCC within up-to-seven criteria and Child–Pugh class A. We excluded patients with HCC exceeding the up-to-seven criteria, HCC with tumor thrombosis and distant metastasis, HCC treated with additional TACE 3 months before or after each TACE, which a single TACE could not cure completely, patients with Child–Pugh class B before undergoing TACE, and patients undergoing anticoagulant therapy (Figure 1). All patients did not have any hematological diseases. The remaining 189 HCC tumors in 114 patients were included in our study. A total of 189 HCC tumors underwent a total of 189 TACEs (66 patients underwent 1st TACE, 15 patients underwent 1st and 2nd TACE, 5 patients underwent 2nd TACE, 10 patients underwent 1st through 3rd TACE, 2 patients underwent 1st and 3rd TACE, 3 patients underwent 1st through 4th TACE, 1 patient underwent 2nd through 4th TACE, 1 patient underwent 3rd and 4th TACE, 1 patient underwent 4th TACE, 1 patient underwent 1st through 6th TACE, 1 patient underwent 2nd through 6th TACE, 1 patient underwent 3rd through 6th TACE, 2 patients underwent 1st, 3rd, 4th, 5th, and 6th TACE, 1 patient underwent 5th and 6th TACE, 1 patient underwent 6th TACE, 1 patient underwent 3rd and 6th TACE, 1 patient underwent 3rd, 4th, and 6th TACE, and 1 patient underwent 8th through 10th TACE.). In other words, each HCC undergoing each TACE was defined as one HCC case. All included HCC were completely cured by a single TACE. We first investigated the change in liver function from Child–Pugh class A to class B in patients with HCC between before and 3 months after each TACE and the risk factors for liver dysfunction in the transition from Child–Pugh class A to class B after each TACE. Next, we studied the risk factors for early recurrence after each TACE. None of the patients had uncontrolled ascites or uncontrolled hepatic encephalopathy at the time of each TACE. Some previous studies [16,17,18] have reported changes in liver function before TACE compared to 3 months after TACE. For this reason, we excluded HCC treated with additional TACE 3 months before or after each TACE and compared changes in liver function between before and 3 months after each TACE. The study was approved by the local ethics committee of Nara Medical University and was performed in accordance with the ethical standards of the Declaration of Helsinki. All participants provided informed consent.

### 2.2. TACE

TACE was conducted using a single femoral approach following the Seldinger technique. Angiography was performed using a microcatheter inserted into the feeding branches, super-selectively. The feeding branches were embolized with a mixture of epirubicin and iodized oil (lipiodol; Laboratoire Andre Guerbet, Aulnay-sous-Bois, France). Collateral artery embolization was performed if branches, such as the phrenic artery and internal thoracic artery, were engaged in the tumor blood supply. In cases with several tumors in the liver, the same TACE was performed on each individual tumor, regardless of the number or location of tumors. Every TACE is considered one session.

### 2.3. Follow-Up

All patients underwent dynamic CT and/or dynamic MRI and blood examination every 2–3 months after undergoing TACE. Radiologic responses to therapy were evaluated according to the modified response evaluation criteria in solid tumors [19]. We focused mainly on tumor staining while diagnosing recurrence.

### 2.4. ALBI Score

The ALBI score was calculated using Alb and T-Bil as follows: ALBI score = (log_10_ T-Bil [μmol/L] × 0.66) + (Alb [g/L] × −0.085) [11,12]. Grades 1, 2, and 3 correspond to ALBI scores of ≤−2.60, <−2.60 to ≤−1.39, and >−1.39, respectively. A modified version has been defined that consists of four grades (1, 2a, 2b, and 3), wherein ALBI grade 2 has been evaluated in more detail based on a cutoff ALBI score of −2.27 for indocyanine green retention rate at 15 min (<30%) [11].

### 2.5. Statistical Analysis

Differences between the groups, which were normally distributed, were analyzed using Student’s *t*-test. Categorical data were analyzed using Fisher’s exact test. Univariate and multivariate analyses were performed to determine independent risk factors for liver dysfunction from Child–Pugh class A to class B after TACE and independent risk factors for early recurrence after TACE using logistic regression analysis. The data are expressed as average ± standard deviation. A two-tailed *p*-value < 0.05 was considered statistically significant. Analyses were conducted using EZR (Saitama Medical Center, Jichi Medical University), which is a graphical user interface for R (The R Foundation for Statistical Computing, version 4.0.3). Specifically, EZR is a modified version of R commander (version 2.7-1) that includes statistical functions frequently used in biostatistics [20].

## 3. Results

### 3.1. Clinical Characteristics of Patients and HCC

Table 1 shows the patient characteristics. The average age of the patients with HCC was 73.7 ± 8.84 years. The study population comprised 88 men and 26 women. Among these, 18 patients had hepatitis B virus, 46 had hepatitis C virus, 26 abused alcohol, 17 had nonalcoholic steatohepatitis, and 7 patients had other conditions. Table 2 presents the characteristics of HCC at the time of undergoing each TACE. The average each TACE trial count for HCC was 2.13 ± 1.66th. The average maximum tumor size was 1.92 ± 0.89 cm. Among HCC, 73 had one tumor, 54 had two tumors, 28 had three tumors, 30 had four tumors, and 4 had five tumors. Of these tumors, 61 were located in both lobes and 128 were located in a single lobe. A total of 116 HCC tumors recurred within 6 months after TACE. The trial count of TACE showed no significant association with the sum of the diameter of the largest tumor (in cm) and the number of tumors (r = 0.0877, *p* = 0.23, according to Spearman’s rank correlation).

### 3.2. Change in Liver Function from Child–Pugh Class A to Class B after TACE

The Alb level was lower in patients with HCC after TACE than before TACE. In contrast, the ALBI scores and Child–Pugh scores were higher in patients with HCC after TACE than before TACE. There was no difference in PT and T-Bil level in patients with HCC between before and after TACE. Overall, 24 patients with HCC (12.7%) changed from Child–Pugh class A to class B after TACE (Table 3).

### 3.3. Risk Factors for Liver Dysfunction from Child–Pugh Class A to Class B after TACE

In the univariate analysis, Alb level, PT, Child–Pugh score, and ALBI score before TACE were associated with liver dysfunction from Child–Pugh class A to class B after TACE (Table 4). To determine the risk factors for liver dysfunction from Child–Pugh class A to class B after TACE, we performed multivariate analysis using Alb level, PT, Child–Pugh score, and ALBI score with *p* < 0.05 in the univariate analysis. PT and ALBI score before TACE were significantly associated with liver dysfunction from Child–Pugh class A to class B after TACE (Table 4). Receiver-operating characteristic (ROC) curve analysis revealed that a cutoff PT of 70% had a specificity of 89.9%, sensitivity of 39.1%, and an area under the curve (AUC) of 0.794, whereas a cutoff ALBI score of −2.27 had a specificity of 78.2%, sensitivity of 83.3%, and an AUC of 0.892. There was no difference in the prediction of liver dysfunction from Child–Pugh class A to class B after TACE between the ROC curve of PT and the ALBI score before TACE. The combination of ALBI score and PT before TACE had a high predictive ability for liver dysfunction from Child–Pugh class A to class B after TACE (Figure 2). The predictive ability had a specificity of 100%, sensitivity of 91.7%, positive predictive value of 100%, and negative predictive value of 98.8%.

### 3.4. Risk Factors for Early Recurrence after TACE

Of the 189 HCC tumors, 116 (61.4%) recurred within 6 months after TACE (Table 2), and patients who underwent repeat TACE had a higher risk of early HCC recurrence than those who underwent initial TACE (Figure 3a). In univariate analysis, the number of tumors and trial counts of TACE were associated with early recurrence after TACE (Table 5). To determine the risk factors for early recurrence after TACE, we performed multivariate analysis using tumor number and trial count of TACE with *p* < 0.05 in univariate analysis. Tumor number and trial counts of TACE were significantly associated with early recurrence after TACE (Table 5). Previous studies have reported that repeat TACE has a high risk of liver dysfunction [5,21]. Therefore, we investigated the relationship between the trial count of TACE and liver dysfunction. Repeat TACE was associated with a higher risk of liver dysfunction after TACE compared to initial TACE (Figure 3b). However, repeat TACE was not significantly associated with a higher risk of liver dysfunction after TACE in the multivariate analysis (Table 4).

## 4. Discussion

Recently, the JSH and European Association for the Study of the Liver have recommended the use of TACE or MTAs for up to ≥4 HCCs [4,22]. TACE is an effective treatment for HCC. However, HCC has a high recurrence rate [7] because of the underlying disease, cirrhosis. Because the liver function of patients with HCC is occasionally decreased following TACE, MTAs should be administered before patients experience this decline in liver function. In fact, in the present study, 12.7% of the patients with HCC within up-to-seven criteria exhibited a decline in liver function from Child–Pugh class A to class B after TACE. Furthermore, previous studies have reported that the treatment efficiency of MTAs is associated with liver function [5,9,23] and that the use of MTAs improves the prognosis of advanced HCC compared with TACE [6].

In the present study, the combination of ALBI score and PT before TACE was determined to be a useful tool for predicting liver dysfunction after TACE in patients with Child–Pugh class A with HCC within up-to-seven criteria. Recent studies have reported that the ALBI score is more effective than the Child–Pugh score for evaluating liver function and that the ALBI score is associated with the prognosis of patients with HCC [14,15,24]. The prevention of the decline in ALBI score after TACE may improve the prognosis of patients with HCC. In fact, recent studies have reported that the ALBI score is associated with the prognosis of patients with HCC undergoing TACE [25,26,27]. In addition, a previous study reported that PT before TACE is a risk factor for liver dysfunction after TACE [28]. Although ALBI score and PT (individually) before TACE are useful biomarkers to predict liver dysfunction in patients with HCC undergoing TACE, it is considered that their predictive ability can be improved by combining them. In Japan and some other countries, MTA is not recommended for patients with HCC with Child–Pugh class B due to side effects such as liver dysfunction and treatment efficiency [4,29]. We believe that a combination of ALBI score and PT can prevent the decline of liver function from Child–Pugh class A to class B following TACE and can expand the treatment options. Notably, the treatment efficiency of MTAs is better in patients with HCC with Child–Pugh class A than class B. Furthermore, previous studies have reported that ALBI score [12,13,30] and PT [31,32] (individually) are prognostic markers for HCC. The combination of ALBI score and PT may be a more useful prognostic marker for HCC than ALBI score and PT individually. Thus, the combination of ALBI score and PT before TACE may serve as a useful tool for predicting liver dysfunction.

In the present study, patients with HCC within up-to-seven criteria as well as those beyond up-to-seven criteria had a high recurrence rate after TACE [7]. A recent study reported that repeat TACE decreased the treatment efficiency [33]. Furthermore, the present study reported that the recurrence of HCC after TACE depends on the number of tumors and trial count of TACE and that the rate of liver dysfunction was higher after repeat TACE than after initial TACE, which corroborate the findings of a previous study [7].

Our present study has several limitations, including its single-center trial design and small sample size. All HCCs were diagnosed using dynamic CT and/or MRI. However, all patients with HCC did not undergo pathological examination for HCC diagnosis. The HCC histological grade might affect the treatment efficiency and the decline in liver function after TACE. Recently, new prognostic tools [4,34,35,36] or new technologies, such as circulating tumor DNA or liquid biopsies [37], for the assessment of HCC have been developed, and the determination of the histological grade of HCC has become easier. We should continue to investigate changes in liver function after TACE in patients with HCC within up-to-seven criteria using new prognostic tools or new technologies. The survival rate for HCC has drastically improved in recent years due to improved diagnostic and treatment methods. We hope that our findings can help to further improve the prognosis of HCC [38]. In summary, 12.7% of the patients with HCC within up-to-seven criteria exhibited a decline in liver function from Child–Pugh class A to class B after TACE. Moreover, the combination of ALBI score and PT before TACE is a useful tool for predicting liver dysfunction after TACE for Child–Pugh class A patients with HCC within up-to-seven criteria. Based on our findings, both the tumor number and trial count of TACE are risk factors for early recurrence after TACE.

## Figures and Tables

**Figure 1 jcm-10-04838-f001:**
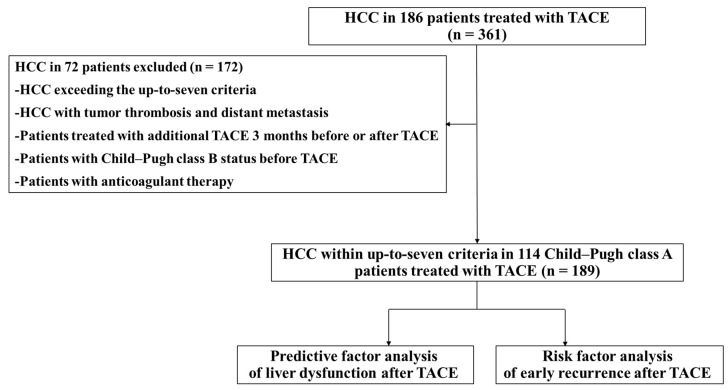
Study design. A total of 361 hepatocellular carcinoma (HCC) tumors in 186 patients were treated with transcatheter arterial chemoembolization (TACE). We excluded patients with HCC exceeding the up-to-seven criteria at the time of each TACE, HCC with tumor thrombosis and distant metastasis at the time of each TACE, additional TACE treatment 3 months before or after undergoing each TACE, Child–Pugh class B status before undergoing each TACE, and patients with anticoagulant therapy. We performed predictive factor analysis of liver dysfunction and risk factor analysis of early recurrence after each TACE in 189 HCC tumors from 114 patients with Child–Pugh class A. HCC, hepatocellular carcinoma; TACE, transcatheter arterial chemoembolization.

**Figure 2 jcm-10-04838-f002:**
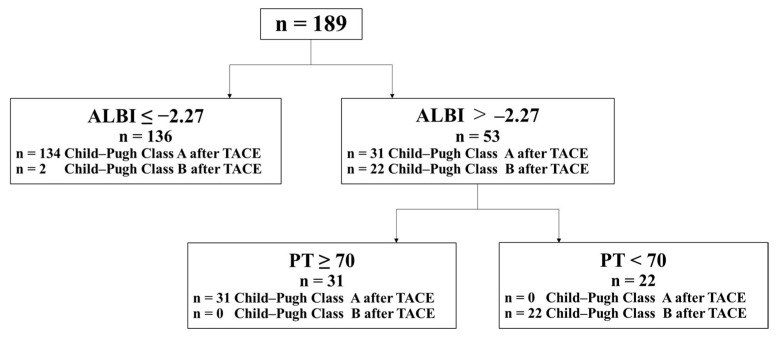
Predicting liver dysfunction after TACE using a combination of ALBI score and PT. The predictive ability of the combination of the ALBI score and PT for liver dysfunction in the transition from Child–Pugh class A to class B after TACE had a specificity of 100%, sensitivity of 91.7%, positive predictive value of 100%, and negative predictive value of 98.8%. ALBI, albumin–bilirubin; PT, prothrombin time; TACE, transcatheter arterial chemoembolization.

**Figure 3 jcm-10-04838-f003:**
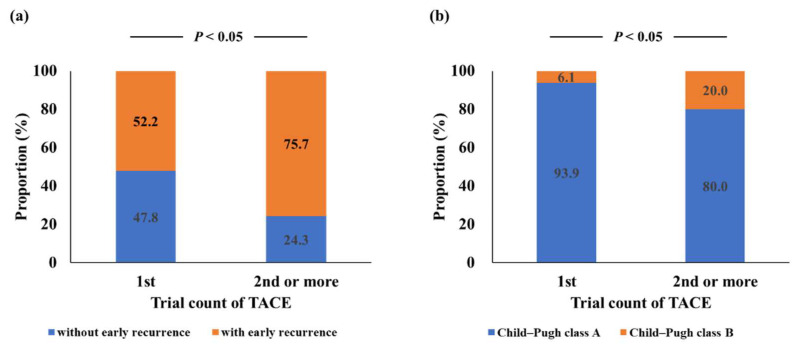
Differences in early recurrence and liver dysfunction rates between initial and repeat TACE. (**a**) Repeat TACE was associated with a higher risk of early recurrence in Child–Pugh class A patients with HCC within the up-to-seven criteria than in those with initial TACE. (**b**) Repeat TACE was associated with a higher risk of liver dysfunction in the transition from Child–Pugh class A to class B than in initial TACE. TACE, transcatheter arterial chemoembolization; HCC, hepatocellular carcinoma.

**Table 1 jcm-10-04838-t001:** Patients’ characteristics.

Variable	
Age (years)	73.7 ± 8.84
Sex (male/female)	88/26
Etiology (HBV/HCV/alcohol/NASH/others)	18/46/26/17/7

HBV, hepatitis B virus; HCV, hepatitis C virus; NASH, nonalcoholic steatohepatitis.

**Table 2 jcm-10-04838-t002:** Characteristics of HCC at the time of each TACE.

Variable	
AFP (ng/mL)	44.2 ± 98.3
Trial count of TACE (1st/2nd/ 3rd and above)	99/36/54
Maximum tumor size (cm)	1.92 ± 0.89
Tumor number (1/2/3/4/5)	73/54/28/30/4
Tumor location (both lobes/one lobe)	61/128
Early recurrence after TACE (+/−)	116/73

AFP, alpha-fetoprotein; HCC, hepatocellular carcinoma; TACE, transcatheter arterial chemoembolization.

**Table 3 jcm-10-04838-t003:** Changes in liver function after TACE.

Variable	Before TACE	3 Months after TACE	*p*
Albumin (g/dL)	3.89 ± 0.499	3.75 ± 0.520	<0.05
Prothrombin time (%)	83.3 ± 13.0	80.5 ± 15.4	NS
Total bilirubin > 1.5 mg/dL	1.00 ± 0.458	1.08 ± 0.540	NS
ALBI score	−2.53 ± 0.45	−2.39 ± 0.48	<0.05
Child–Pugh score	5.29 ± 0.453	5.59 ± 0.824	<0.05
Child–Pugh class A/B	189/0	165/24	<0.05

ALBI score, albumin–bilirubin score; TACE, transcatheter arterial chemoembolization; NS, nonsignificant.

**Table 4 jcm-10-04838-t004:** Risk factors for liver dysfunction after TACE.

	Univariate Analysis		Multivariate Analysis	
Variable	OR (95% CI)	*p*	OR (95% CI)	*p*
Age > 70 years old	0.541 (0.224–1.30)	0.171		
Sex (male vs. female)	0.393 (0.112–1.38)	0.145		
Albumin < 3.5 g/dL	9.79 (3.82–25.1)	<0.05	4.34 (0.716–26.3)	0.110
Prothrombin time < 70%	5.39 (1.96–14.8)	<0.05	12.4 (1.84–83.4)	<0.05
Total bilirubin > 2 mg/dL	4.91 (0.777–31.0)	0.0908		
Child–Pugh score 6 vs. 5	18.6 (5.96–57.9)	<0.05	1.32 (0.254–6.87)	0.741
ALBI score > −2.27	19.3 (6.17–60.1)	<0.05	9.08 (1.93–42.8)	<0.05
AFP > 20 ng/mL	1.64 (0.684–3.95)	0.267		
Maximum tumor size > 3 cm	0.73 (0.203–2.62)	0.630		
Tumor number > 3	1.56 (0.571–4.28)	0.385		
Tumor location (both lobes/one lobe)	1.95 (0.816–4.64)	0.133		
Trial count of TACE ≥ 2	0.943 (0.329–2.70)	0.913		

ALBI score, albumin–bilirubin score; AFP, alpha-fetoprotein; CI, confidence interval; OR, odds ratio; TACE, transcatheter arterial chemoembolization.

**Table 5 jcm-10-04838-t005:** Risk factors for early recurrence after TACE.

	Univariate Analysis		Multivariate Analysis	
Variable	OR (95% CI)	*p*	OR (95% CI)	*p*
Age > 70 years old	0.839 (0.439–1.60)	0.594		
Sex (male vs. female)	0.715 (0.366–1.40)	0.326		
Albumin < 3.5 g/dL	0.999 (0.500–2.00)	0.999		
Prothrombin time < 70%	1.12 (0.443–2.81)	0.817		
Total bilirubin > 2 mg/dL	2.57 (0.282–23.5)	0.403		
Child–Pugh score 6 vs. 5	1.28 (0.665–2.46)	0.461		
ALBI score > −2.27	1.23 (0.637–2.37)	0.539		
AFP > 20 ng/mL	1.73 (0.902–3.33)	0.0985		
Maximum tumor size > 3 cm	0.793 (0.360–1.75)	0.564		
Tumor number > 3	4.74 (1.75–12.9)	<0.05	4.73 (1.73–13.0)	<0.05
Tumor location (both lobes/one lobe)	2.01 (1.04–3.89)	0.077		
Trial count of TACE ≥ 2	2.71 (1.21–6.08)	<0.05	2.70 (1.18–6.16)	<0.05

ALBI score, albumin–bilirubin score; AFP, alpha-fetoprotein; CI, confidence interval; OR, odds ratio; TACE, transcatheter arterial chemoembolization.

## Data Availability

Informed consent for data sharing was not obtained but the presented data are anonymized and the risk of identification is low.

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
