# Peer review of "The Combination of Albumin–Bilirubin Score and Prothrombin Time Is a Useful Tool for Predicting Liver Dysfunction after Transcatheter Arterial Chemoembolization in Child–Pugh Class A Patients with Hepatocellular Carcinoma within Up-to-Seven Criteria"

_jcm, 2021, doi:10.3390/jcm10214838_

Round 1

Reviewer 1 Report

In this retrospective study the authors assessed patients with HCC beyond up-to-seven criteria, treated with transcatheter arterial chemoembolization (TACE); In particular, the study was focused on liver function deterioration after TACE in patients with HCC within up-to-seven.

They studied 189 HCC tumors within up-to-seven criteria in 114 Child–Pugh class A patients. They found that twenty-four (12.7%) patients were changed from Child–Pugh class A to B after TACE, 31 and 116 (61.4%) tumors exhibited recurrence within 6 months after TACE. Prothrombin time (PT) 32 and albumin–bilirubin (ALBI) score before TACE were significantly associated with liver dysfunction from Child–Pugh class A to B. The combination of PT and ALBI score before TACE had high predictive ability for liver dysfunction from Child–Pugh class A to B after TACE (specificity = 100%, 35 sensitivity = 91.7%). They found that combined use of pre-TACE PT and ALBI score had a high predictive ability for liver dysfunction after TACE for Child–Pugh class A patients with HCC within up-to-seven criteria.

The study is of interest and of potential clinical relevance, however some points deserve further details and should be addressed.

-Study design and patients: 186 patients within up-to-seven criteria and Child–Pugh class A who underwent TACE between March 2015 and August 2020 were studied. Please, describe in details the TACE procedure: patients with different tumor number (1/2/3/4/5) and Tumor location (both lobes/one lobe) how were differently treated? One or more TACE sessions? How much arterial vessel embolization? This is a relevant issue since the number of embolized vessel might affect liver function deterioration as a result of more extensive liver parenchyma ischemic damamge.

-In discussion, the authors should more emphasize the clinical relevance of liver function deterioration (as result of transition Child-Pugh A to Child-Pugh B class) in determining less treatment possibility (especially for systemic treatments that require only Child-Pugh class A patients), as extensively described and reported in a comprehensive review article (Non-transplant therapies for patients with hepatocellular carcinoma and Child-Pugh-Turcotte class B cirrhosis. Lancet Oncol. 2017 Feb;18(2):e101-e112.).

-I would also suggest stressing in the discussion that according to study results, the selection of HCC patients candidate for TACE could be improved thus reducing the risk of liver function deterioration to Child-Pugh B class for which presently only limited systemic treatments have provide treatment efficacy (Metronomic capecitabine as second-line treatment in hepatocellular carcinoma after sorafenib failure. Dig Liver Dis. 2015 Jun;47(6):518-22; Metronomic capecitabine vs. best supportive care in Child-Pugh B hepatocellular carcinoma: a proof of concept. Sci Rep. 2018 Jul 3;8(1):9997.)  

-Although there are few data available treatment of patients within the up-to-seven criteria, to further improve the introduction I would suggest to recall the impact of the changes in the aetiological and diagnostic management of HCC patients during the last 15 years significantly improving outcome of loco-regional treatments, as recently reported (The evolutionary scenario of hepatocellular carcinoma in Italy: an update. Liver Int. 2017 Feb;37(2):259-270).

Author Response

We thank the reviewers for their careful review of our manuscript and for their useful feedback. To address their comments, we have thoroughly revised the manuscript. Please find our point-by-point responses to each of the reviewers’ comments below. In addition, the revised manuscript has also undergone careful proofreading by an English language editor.

Response to Reviewer 1

Thank you for reviewing our manuscript. Our point-by-point responses to your comments are as follows:

1) -Study design and patients: 186 patients within up-to-seven criteria and Child–Pugh class A who underwent TACE between March 2015 and August 2020 were studied. Please, describe in details the TACE procedure: patients with different tumor number (1/2/3/4/5) and Tumor location (both lobes/one lobe) how were differently treated? One or more TACE sessions? How much arterial vessel embolization? This is a relevant issue since the number of embolized vessel might affect liver function deterioration as a result of more extensive liver parenchyma ischemic damamge.

Response to 1)

Thank you for this valuable comment. We have described the TACE procedure in the METHODS as follows:

“TACE was conducted using a single femoral approach following the Seldinger technique. Angiography was performed using a microcatheter inserted into the feeding branches, super-selectively. The feeding branches were embolized with a mixture of epirubicin and iodized oil (lipiodol; Laboratoire Andre Guerbet, Aulnay-sous-Bois, France). Collateral artery embolization was performed if branches, such as the phrenic artery and internal thoracic artery, were engaged in the tumor blood supply. In cases with several tumors in the liver, the same TACE was performed on each individual tumor, regardless of the number or location of tumors. Every TACE is considered one session.”

2)-In discussion, the authors should more emphasize the clinical relevance of liver function deterioration (as result of transition Child-Pugh A to Child-Pugh B class) in determining less treatment possibility (especially for systemic treatments that require only Child-Pugh class A patients), as extensively described and reported in a comprehensive review article.

Response to 2)

Thank you for this valuable comment. In the revised DISCUSSION section, we state that:

“In Japan and some other countries, MTA is not recommended for patients with HCC with Child–Pugh class B due to side effects such as liver dysfunction and treatment efficiency. We believe that a combination of ALBI score and PT can prevent the decline of liver function from Child–Pugh class A to class B following TACE and can expand the treatment options.”

3)-I would also suggest stressing in the discussion that according to study results, the selection of HCC patients candidate for TACE could be improved thus reducing the risk of liver function deterioration to Child-Pugh B class for which presently only limited systemic treatments have provide treatment efficacy.

Response to 3)

Thank you for this valuable comment. In the revised DISCUSSION section, we state:

“We believe that a combination of ALBI score and PT can prevent the decline of liver function from Child–Pugh class A to class B following TACE and expand the treatment options. Notably, the treatment efficiency of MTAs is better in patients with HCC with Child–Pugh class A than class B.”

4)-Although there are few data available treatment of patients within the up-to-seven criteria, to further improve the introduction I would suggest to recall the impact of the changes in the aetiological and diagnostic management of HCC patients during the last 15 years significantly improving outcome of loco-regional treatments, as recently reported.

Response to 4)

Thank you for this valuable comment. We have revised the DISCUSSION section to state that:

“The survival rate for HCC has drastically improved in recent years due to improved diagnostic and treatment methods. We hope that our findings can help to further improve the prognosis of HCC.”

Reviewer 2 Report

Dear authors, in your paper you try to identify predictive factors of liver function deterioration (Transition from Child Pugh A to B) after TACE in patients with Child Pugh A cirrhosis and HCC within Up-to-seven criteria. You also try to identify predictive factors for tumor recurrence after TACE.

You retrospectively evaluate a cohort of 189 HCC tumors within up-to-seven criteria treated with TACE. 24 (12,7%) tumors changed from Child Pugh A to B. 116 (61,4%) tumors presented recurrence < 6 months.

ALBI score and prothrombin time were independent predictors of liver function deterioration. The combination of ALBI and PT showed a 100% specificity and 91.7% sensitivity for Child Pugh deterioration after TACE.

Factors associated with early HCC recurrence (<6 months) after TACE were number of nodules and repeated TACE procedure.

The conclusion is that ALBI + PT is a useful tool for predicting Child Pugh deterioration after TACE (better than Child Pugh alone). However you also state that these patients should be treated with molecular targeted agents, (which in my opinion is speculative and not supported by the data) and that repeated TACE was associated with a higher risk of liver dysfunction which is not supported by the data (see comments).

There are some issues that require clarification:

  • The terminology is confusing. I had to read the paper several times to clearly understand that what you are calling 189 HCC tumors refers to 189 TACE procedures. It seems that each HCC lesion is being considered as a case, and the fact that the description refers to HCC tumors or patients indistinctly can be misleading. Please clarify if you are considering each TACE procedure as a case, and how many patients are repeated cases.
  • The TACE protocol (drug, embolizing agent, time schedule for repeated TACE) should be explained, and the reason for excluding cases treated with additional TACE 3 months before or after. Response to initial TACE and change in the liver function, may modify the decision of repeating procedures and the time interval. Excluding some patients or some TACE procedures may induce a selection bias. Patients that receive a second TACE within the next 3 months are patients that have not deteriorated their liver function (otherwise would not have undergone another TACE). Exclusion of these patients may overestimate the number of patients that deteriorate liver function after TACE.
  • In the methods section you mention that patients were evaluated with mRECIST criteria but radiological response to TACE is not reported. I assume you are considering recurrence when tumor appears again after a complete response, but 100% complete responses after TACE is difficult to obtain. A better description of the radiological response and what is considered recurrence or progression should be given.
  • In the case of repeated TACE for a same patient, I am not sure that a single patient may be included as 2 different cases with different outcome regarding recurrence/progression. Progression is a time-dependent variable and also influences the decision of additional TACE procedures. If a patient is treated with repeated TACE until progression, last TACE will always be closer to the progression than the first one.
  • In the “Risk factors for early recurrence after TACE” section you mention that repeat TACE was associated with a higher risk of liver disfunction after TACE, however in the previous section which analyzes specifically the risk factors for liver disfunction you evaluate this parameter and show that it is not associated with liver disfunction (Table 4). Please clarify.
  • Conclusions should be limited to what is explored in the paper. Several statements are not supported by the data and the message should be tempered. In different paragraphs appear that patients predicted to experience Child Pugh deterioration after TACE should be treated with molecular targeted agents, or that patients within up-to-seven criteria should not be treated with repeated TACE. The fact that some patients experience deterioration in Child Pugh status after TACE in this study does not imply that these patients should be better treated with molecular targeted agents from the beginning. Some patients experience deterioration of liver function after molecular targeted agents, and ALBI has also been associated with prognosis and liver function deteriorarion in that scenario. There are also several factors that may influence liver function deterioration that are not specified in the paper (alcohol consumption, HCV status, dose of the drug administered and extension of embolization, infections, etc…). Moreover deterioration of liver function may be transitory and there is no mention regarding prognosis of these patients in the paper (describing a shorter survival for these patients compared to those who did not present liver function deterioration may reinforce your message). With all of these uncertainties such recommendations for using molecular targeted agents in these patients cannot be made until a randomized clinical trial shows a benefit for that approach. The message should be tempered, and just suggest that hypothetical scenario, or the use of ALBI and PT for selection of patients for a clinical trial.

Author Response

We thank the reviewers for their careful review of our manuscript and for their useful feedback. To address their comments, we have thoroughly revised the manuscript. Please find our point-by-point responses to each of the reviewers’ comments below. In addition, the revised manuscript has also undergone careful proofreading by an English language editor.

Response to Reviewer 2

Thank you for reviewing our manuscript. Our point-by-point responses to your comments are as follows:

1)The terminology is confusing. I had to read the paper several times to clearly understand that what you are calling 189 HCC tumors refers to 189 TACE procedures. It seems that each HCC lesion is being considered as a case, and the fact that the description refers to HCC tumors or patients indistinctly can be misleading. Please clarify if you are considering each TACE procedure as a case, and how many patients are repeated cases.

Response to 1)

Thank you for pointing this out. We have revised the Study design and patients subsection of the METHODS to state that:

“A total of 189 HCC tumors underwent a total of 189 TACEs (66 patients underwent 1st TACE, 15 patients underwent 1st and 2nd TACE, 5 patients underwent 2nd TACE, 10 patients underwent 1st through 3rd TACE, 2 patients underwent 1st and 3rd TACE, 3 patients underwent 1st through 4th TACE, 1 patient underwent 2nd through 4th TACE, 1 patient underwent 3rd and 4th TACE, 1 patient underwent 4th TACE, 1 patient underwent 1st through 6th TACE, 1 patient underwent 2nd through 6th TACE, 1 patient underwent 3rd through 6th TACE, 2 patients underwent 1st, 3rd, 4th, 5th, and 6th TACE, 1 patient underwent 5th and 6th TACE, 1 patient underwent 6th TACE, 1 patient underwent 3rd and 6th TACE, 1 patient underwent 3rd, 4th and 6th TACE, and 1 patient underwent 8th through 10th TACE.). In other words, each HCC undergoing each TACE was defined as one HCC case. All included HCC were completely cured by a single TACE.”

2)The TACE protocol (drug, embolizing agent, time schedule for repeated TACE) should be explained, and the reason for excluding cases treated with additional TACE 3 months before or after. Response to initial TACE and change in the liver function, may modify the decision of repeating procedures and the time interval. Excluding some patients or some TACE procedures may induce a selection bias. Patients that receive a second TACE within the next 3 months are patients that have not deteriorated their liver function (otherwise would not have undergone another TACE). Exclusion of these patients may overestimate the number of patients that deteriorate liver function after TACE.

Response to 2)

Thank you for these valuable comments. We have revised the manuscript to clarify that:

“TACE was conducted using a single femoral approach following the Seldinger technique. Angiography was performed using a microcatheter inserted into the feeding branches, super-selectively. The feeding branches were embolized with a mixture of epirubicin and iodized oil (lipiodol; Laboratoire Andre Guerbet, Aulnay-sous-Bois, France). Collateral artery embolization was performed if branches, such as the phrenic artery and internal thoracic artery, were engaged in the tumor blood supply. In cases with several tumors in the liver, the same TACE was performed on each individual tumor, regardless of the number or location of tumors. Every TACE is considered one session.” (Study design and patients, MATERIALS AND METHODS).

In addition, we have clarified that:

“We excluded patients with HCC exceeding the up-to-seven criteria, HCC with tumor thrombosis and distant metastasis, HCC treated with additional TACE 3 months before or after each TACE, which a single TACE could not cure completely…” (Study design and patients, MATERIALS AND METHODS).

Finally, we have revised the text to indicate the following:

“Some previous studies have reported changes in liver function before TACE compared to 3 months after TACE. For this reason, we excluded HCC treated with additional TACE 3 months before or after each TACE and compared changes in liver function between before and 3 months after each TACE.” (Study design and patients, MATERIALS AND METHODS).

3)In the methods section you mention that patients were evaluated with mRECIST criteria but radiological response to TACE is not reported. I assume you are considering recurrence when tumor appears again after a complete response, but 100% complete responses after TACE is difficult to obtain. A better description of the radiological response and what is considered recurrence or progression should be given.

Response to 3)

Thank you for your valuable comment. We indicated that “we focused mainly on tumor staining while diagnosing recurrence” in the Follow-up section (MATERIALS AND METHODS).

4)In the case of repeated TACE for a same patient, I am not sure that a single patient may be included as 2 different cases with different outcome regarding recurrence/progression. Progression is a time-dependent variable and also influences the decision of additional TACE procedures. If a patient is treated with repeated TACE until progression, last TACE will always be closer to the progression than the first one.

Response to 4)

Thank you for your valuable comment. We have revised the RESULTs section to state:

“The trial count of TACE showed no significant association with the sum of the diameter of the largest tumor (in cm) and the number of tumors (r = 0.0877, p = 0.23, according to Spearman’s rank correlation).”

5)In the “Risk factors for early recurrence after TACE” section you mention that repeat TACE was associated with a higher risk of liver disfunction after TACE, however in the previous section which analyzes specifically the risk factors for liver disfunction you evaluate this parameter and show that it is not associated with liver disfunction (Table 4). Please clarify.

Response to 5)

Thank you for pointing out this error. We have corrected the RESULTS section as follows:

“Previous studies have reported that repeat TACE has a high risk of liver dysfunction. Therefore, we investigated the relationship between the trial count of TACE and liver dysfunction. Repeat TACE was associated with a higher risk of liver dysfunction after TACE compared to initial TACE (Fig. 3b). However, repeat TACE was not significantly associated with a higher risk of liver dysfunction after TACE in the multivariate analysis (Table 4).”

6)Conclusions should be limited to what is explored in the paper. Several statements are not supported by the data and the message should be tempered. In different paragraphs appear that patients predicted to experience Child Pugh deterioration after TACE should be treated with molecular targeted agents, or that patients within up-to-seven criteria should not be treated with repeated TACE. The fact that some patients experience deterioration in Child Pugh status after TACE in this study does not imply that these patients should be better treated with molecular targeted agents from the beginning. Some patients experience deterioration of liver function after molecular targeted agents, and ALBI has also been associated with prognosis and liver function deteriorarion in that scenario. There are also several factors that may influence liver function deterioration that are not specified in the paper (alcohol consumption, HCV status, dose of the drug administered and extension of embolization, infections, etc…). Moreover deterioration of liver function may be transitory and there is no mention regarding prognosis of these patients in the paper (describing a shorter survival for these patients compared to those who did not present liver function deterioration may reinforce your message). With all of these uncertainties such recommendations for using molecular targeted agents in these patients cannot be made until a randomized clinical trial shows a benefit for that approach. The message should be tempered, and just suggest that hypothetical scenario, or the use of ALBI and PT for selection of patients for a clinical trial.

Response to 6)

Thank you for your valuable comment. We have deleted the following sentences stating that “the patients with HCC undergoing TACE predicted to experience liver function decline after TACE should be treated using MTAs” from the DISCUSSION (second paragraph) and “Some patients with HCC within up-to-seven criteria should not be treated with repeat TACE and should be treated with MTAs because of the high recurrence rate of TACE, the decline in liver function associated with repeat TACE, and the fact that treatment with MTAs maintains the liver function of patients with HCC” from the DISCUSSION (third paragraph).

Round 2

Reviewer 1 Report

The authors addressed the raised points.

Author Response

Thank you for reviewing our manuscript. We appreciate your acceptance of our manuscript.

Reviewer 2 Report

Dear editor, the authors have included descriptions of important design issues, and have moderated the message of the study that is limited to the capacity of ALBI and PT to predict liver disfunction after TACE and the identification of an association between the number of HCC nodules and TACE procedures with early recurrence.

With the authors explanations I have confirmed what I suspected, that several patients are considered more than once and that raises doubts about the capacity of predicting recurrence. For instance the risk of recurrence of a patient with a single 2 cm HCC treated with TACE cannot be compared to a patient that has already been treated a single 5 cm HCC that presents a recurrence in form of a single 2 cm HCC that is treated in a second TACE. The variable trial count of TACE may not reflect all the shades of the disease.

Patients without a complete response are excluded (which may be suitable for the main objective of predicting liver disfunction after TACE).

Speculations regarding prognosis are made but survival is not reported.

However at least in the present version the design is better described and the reader may extract its own conclusions.   

There is an important issue that has not been corrected:

In the section evaluating risk factors for early recurrence after TACE, again, they mention that repeated TACE was associated with a higher risk of liver dysfunction after TACE in the univariate analysis. However in table 4 the OR (95% CI) is 0.943 (0.329-2.70), p = 0.913. This statement has not been completely removed from the Discussion.

Author Response

Thank you for reviewing our manuscript. Our response to your comment is as follow:

1)In the section evaluating risk factors for early recurrence after TACE, again, they mention that repeated TACE was associated with a higher risk of liver dysfunction after TACE in the univariate analysis. However in table 4 the OR (95% CI) is 0.943 (0.329-2.70), p = 0.913. This statement has not been completely removed from the Discussion.

Response to 1)

Thank you for this valuable comment.

We have deleted the following sentences stating that “TACE had a higher risk of liver dysfunction than initial TACE” from the DISCUSSION (third paragraph).